# Can Lycopene Impact the Androgen Axis in Prostate Cancer?: A Systematic Review of Cell Culture and Animal Studies

**DOI:** 10.3390/nu11030633

**Published:** 2019-03-15

**Authors:** Catherine C. Applegate, Joe L. Rowles, John W. Erdman

**Affiliations:** 1Division of Nutritional Sciences, University of Illinois at Urbana-Champaign, Urbana, IL 61801, USA; cca2@illinois.edu (C.C.A.); jrowles2@illinois.edu (J.L.R.); 2Department of Food Science and Human Nutrition, University of Illinois at Urbana-Champaign, Urbana, IL 61801, USA

**Keywords:** prostate cancer, tomato, lycopene, androgen, cell culture, animal

## Abstract

First-line therapy for advanced or metastatic prostate cancer (PCa) involves the removal of tumor-promoting androgens by androgen deprivation therapy (ADT), resulting in transient tumor regression. Recurrent disease is attributed to tumor adaptation to survive, despite lower circulating androgen concentrations, making the blockage of downstream androgen signaling a chemotherapeutic goal for PCa. Dietary intake of tomato and its predominant carotenoid, lycopene, reduce the risk for PCa, and preclinical studies have shown promising results that tomato and lycopene can inhibit androgen signaling in normal prostate tissue. The goal of this systematic review was to evaluate whether mechanistic evidence exists to support the hypothesis that tomato or lycopene interact with the androgen axis in PCa. Eighteen studies (*n* = 5 in vivo; *n* = 13 in vitro) were included in the final review. A formal meta-analysis was not feasible due to variability of the data; however, the overall estimated directions of effect for the compared studies were visually represented by albatross plots. All studies demonstrated either null or, more commonly, inhibitory effects of tomato or lycopene treatment on androgen-related outcomes. Strong mechanistic evidence was unable to be ascertained, but tomato and lycopene treatment appears to down-regulate androgen metabolism and signaling in PCa.

## 1. Introduction

Despite average annual declines in incidence, prostate cancer (PCa) remains the most commonly diagnosed male cancer in the United States, with an estimated three million men currently living with PCa [1]. It is well-understood that primary PCa growth is strongly dependent upon the activity of androgens within the prostate gland, as evidenced by the observed rise of androgen-regulated prostate-specific antigen (PSA) in the serum of men diagnosed with PCa [2]. First-line therapy for advanced or metastatic disease involves androgen suppression through androgen deprivation therapy (ADT) [3]. ADT results in castrate levels of androgens in the bloodstream and subsequent success of initial tumor regression; however, the return of castration-resistant disease inevitably occurs within a few years after ADT and is thought to be the result of adaptive or persistent intratumoral androgen production, metabolism, and signaling [4]. While the mechanisms of androgen metabolism and signaling leading to prostate carcinogenesis and continued tumor growth are still under investigation, the blockade of androgen signaling, in addition to ADT, has been identified as a target goal for chemotherapy.

Abundant epidemiological evidence indicates that tomato consumption and blood levels of the predominant carotenoid found in tomatoes, lycopene, are inversely associated with PCa risk [5,6,7,8,9]. Additional evidence suggests that tomato and lycopene interact with the androgen axis to reduce blood levels of PSA [10,11], as well as reduce the risk of advanced stage, lethal PCa [8,12,13]. Animal and cell culture studies reveal an interaction between lycopene and androgen status and signaling, further indicating a potential protective role of tomato and lycopene intake for PCa patients.

Androgens, such as testosterone and dihydrotestosterone (DHT), are male sex hormones required for prostate differentiation and the maintenance of prostate structure and function throughout the lifespan [14]. Once delivered to the prostate from the testes via the bloodstream, androgens can either be converted to more active forms or metabolized to less active forms by a variety of hydroxysteroid dehydrogenase (HSD) enzymes. For example, testosterone is a potent ligand for the androgen receptor (AR), but is converted to DHT by two isoforms of 5-α-reductase (SRD5A1 and SRD5A2). DHT has a higher affinity for binding to the AR, which leads to AR nuclear translocation, DNA binding, and the transcription of genes related to growth and survival pathways [15].

Our laboratory has previously shown that castrated male F344 rats accumulated two times more lycopene in the liver than intact rats or castrated rats treated with testosterone repletion [16,17]. We have also shown that short-term tomato or tomato carotenoid feeding led to significant decreases of serum testosterone in rats, with carotenoid intake interacting with castration to further decrease serum testosterone [18]. In addition, both castration and carotenoid intake resulted in the regulation of prostatic androgen-related enzyme gene expression. Expression of HSD17β4 was significantly higher after castration, as well as between castrated rats fed tomato or lycopene diets when compared to castrated or intact, control-fed rats. HSD17β4 activity results in the metabolism of more potent androgens to less potent forms, and HSD17β4 silencing has been shown to increase AR nuclear localization and PSA expression [19]. This upregulation of HSD17β4 may indicate a switch from androgen signaling propagation to androgen deactivation. Modulation of androgen-related enzyme gene expression by tomato and lycopene is supported by the observed upregulation of HSD17β4 and downregulation of SRD5A2 in the prostate of Copenhagen rats supplemented with lycopene [20]. Supplementation with lycopene also decreased the prostatic expression of steroid target genes prostatic steroid binding chains C1 and C3, cystatin-related protein 2, and seminal vesicle secretion protein IV. In addition, lycopene supplementation of human primary prostatic epithelial cells (PrE) reduced the expression of AR chaperone heat shock protein 90 (HSP90) and protein DJ1, a positive regulator of AR-dependent transcription [21].

These results support the hypothesis that lycopene metabolism is affected by androgen status and that tomato and lycopene interact with the androgen axis in the normal prostate by modulating the expression of genes involved in androgen metabolism and signaling. As such, we hypothesized that tomato and lycopene could similarly interact with the androgen axis during PCa. Because interference with androgen signaling is a critical chemotherapeutic goal for PCa treatment, the primary objective of this review was to systematically evaluate whether mechanistic evidence exists to support a role for tomato or lycopene interaction with the androgen axis during PCa. To accomplish this objective, we included animal and cell culture studies exploring this relationship and evaluated the overall strength and comparability of the evidence. This study is novel in that there is a general dearth of systematic reviews of animal and cell culture studies, and, to our knowledge, no studies currently exist to mechanistically evaluate the relationship between tomato or lycopene and the androgen axis during PCa. While strong mechanistic evidence was unable to be ascertained, the results showed that tomato and lycopene appeared to down-regulate androgen metabolism and signaling in PCa tissues.

## 2. Materials and Methods

### 2.1. Study Selection Criteria

Currently, no validated guidelines or tools exist for conducting systematic reviews or for evaluating the validity and quality of mechanistic studies. As part of an effort to utilize a cohesive and standardized set of guidelines for systematically reviewing evidence from cell culture and animal studies, this systematic review was conducted in accordance with the framework outlined by the World Cancer Research Fund (WCRF) International/University of Bristol (UoB) [22]. In addition, care was taken to follow PRISMA reporting guidelines as closely as possible [23].

Cell culture and animal studies that met the following criteria were included in this systematic review: (a) evaluated the relationship between tomatoes and/or their primary bioactive, lycopene, and androgen metabolism, androgen signaling, or androgen-mediated outcomes in PCa through cell culture studies evaluating direct androgen endpoints in PCa cell lines, cell culture studies comparing androgen-sensitive versus androgen-insensitive PCa cell lines, or animal studies evaluating direct androgen endpoints in experimental animal models of PCa (carcinogen-induced, xenograft, transplantable, or transgenic); (b) methodology was documented in replicable detail; (c) used whole tomato, tomato extract, or lycopene as a single supplement; (d) were written in English; and (e) were peer-reviewed original research articles or theses.

### 2.2. Literature Search

We conducted a comprehensive literature search of PubMed, Web of Science, and the Cochrane Library using a combination of the following keywords and their variants: tomato, lycopene, testosterone, androgen, dihydrotestosterone, DHT, prostate specific antigen, PSA, prostate cancer, and prostate neoplasm (up to 23 January 2019). Titles and abstracts of articles that were identified by the search results were screened against the study selection criteria. Full texts of potentially relevant articles identified by abstract screening were further reviewed for study inclusion or exclusion. To minimize the risk of excluding potentially relevant studies, we also conducted a reference list search (i.e., backward search) and cited reference search (i.e., forward search) from studies meeting the study selection criteria. Studies identified through this process were further screened and evaluated using the afore-mentioned criteria. We repeated reference searches on all newly identified studies until no further relevant studies were found. Two authors (CCA and JLR3) individually determined the inclusion/exclusion of all studies retrieved in full text, and discrepancies were resolved through discussion.

### 2.3. Data Extraction and Quality Assessment

Data extraction was performed according to study type (animal or cell culture) using the recommendations set forth by the WCRF/UoB framework as a guide [22]. The following information was extracted from each animal study: animal model, housing conditions and dietary information for experimental and control groups, experimental design (investigator blinding, randomization or grouping of animals, etc.), duration of follow-up, androgen-related outcomes analyzed, results of androgen-related outcomes, sample size, and *p*-values. The following information was extracted from each cell culture study: names of cell lines, whether cell lines were established patient-derived tumor cell lines or freshly isolated primary cells, whether cell lines were authenticated, culture conditions, treatment regime (dose and length of treatment), details of laboratory procedures, outcomes analyzed, results, sample size, and *p*-values.

There is a lack of validated quality assessment (QA) tools to evaluate the risk of bias associated with animal and cell culture studies. QA of animal studies was performed using the SYstematic Review Centre for Laboratory animal Experimentation (SYRCLE) risk of bias tool [24] adapted from the established and validated Cochrane tool [25] for human study risk of bias assessment. Risk of bias was determined to be “high,” “low,” or “unclear.” Total scores were not evaluated using the SYRCLE tool to avoid inappropriate weighting of each category. QA of cell culture studies was performed using the criteria recommended by the WCRF/UoB framework (score range 0–6; a score of 0 was assigned for each parameter not fulfilled or not reported) [22]. Based on the score, studies were rated as low (0–2), moderate (3–4), or high (5–6) quality. QA scores were utilized to provide a measure of the strength of the evidence and to determine if a risk of bias was present for each study, but were not used to determine the inclusion of studies. QA scores were considered when making conclusions about whether the included studies supported the biological plausibility of the causal pathway being investigated.

### 2.4. Albatross Plot Generation

The extreme degree of variation between the methodologies and outcome measures of animal and cell culture studies prevented statistical analysis via a meta-analysis. Due to this variation, outcome measures were grouped according to indirect or direct androgen-related outcomes. In lieu of a meta-analysis, albatross plots for each outcome measure were generated to both graphically represent the data and create an effect estimate for each outcome category measured. An albatross plot, as described by Harrison et al. [26], scatters the *p*-values of each study according to their sample size and according to the observed direction of the effect (positive or negative). In the absence of exact *p*-values provided, the most conservative *p*-value was assigned to that outcome (e.g., if given *p* < 0.05, set *p* = 0.05; if no *p*-value given for a non-significant association [e.g., *p* > 0.05], set *p* = 1). If studies included multiple *p*-values for different outcomes, all *p*-values were included as separate data points. The contour lines extending over the plots represent estimated effect sizes (represented as standardized mean differences [SMD]) to allow for the estimation of the magnitude of treatment effects for individual studies and for their association as a whole. Visual inspection of the albatross plots was used to determine an overall estimated effect for each outcome. Because study outcomes were grouped according to indirect or direct androgen-related outcomes and albatross plots only provide an estimate of the effect of tomato or lycopene for these categories, the results provided by the plots are representative of this estimate, rather than a true statistical analysis. Albatross plots were generated using STATA/IC V14.2 (StataCorp LP, College Station, TX, USA).

## 3. Results

### 3.1. Literature Search

In total, 326 studies were identified from the library search engines. After duplicate removal and adding studies identified from reference lists, 192 studies remained for abstract screening. Subsequently, 28 studies were found to contain potentially relevant information to be further evaluated by full text review. Inclusion was determined according to the afore-mentioned inclusion criteria, with a resulting 18 studies included in the final review. Ten studies were excluded because they did not discuss PCa (*n* = 2), did not evaluate direct androgen outcomes or compare androgen-sensitive vs. androgen-insensitive PCa cell lines (*n* = 7), or did not evaluate lycopene as a single supplement (*n* = 1). Of the 18 included studies, five were animal studies [27,28,29,30,31] and 13 were cell culture studies [32,33,34,35,36,37,38,39,40,41,42,43,44] (Figure 1).

### 3.2. Study Characteristics

Of the five animal studies, three studies utilized rats (transplantable tumor models) [27,29,30] and two studies used mice (*n* = 1 xenograft model [28] and *n* = 1 transgenic model [31]). Animal study characteristics and results are summarized in Table 1. Of the 13 cell culture studies, 11 [32,33,34,36,37,38,39,40,42,43,44] used patient-derived PCa tumor cell lines, one [35] used primary PCa tumor cell lines, and one [41] used rat-derived PCa tumor cell lines. The cell culture studies were further stratified according to lycopene interaction with the androgen axis, as follows: (a) nine studies [32,33,34,36,40,41,42,43,44] evaluated androgen-related outcomes pertaining to indirect lycopene interaction with the androgen axis by comparing differential effects of lycopene between androgen-sensitive (AS) and androgen-insensitive (AI) PCa cell lines (indirect androgen outcomes); and (b) eight studies [32,33,34,35,36,37,38,39] evaluated androgen-related outcomes pertaining to direct lycopene interaction with androgen signaling, androgen metabolism, or androgen-regulated gene expression (direct androgen outcomes). Due to a range of indirect androgen outcomes reported, this group was further subdivided into studies that measured (i) growth [32,33,36,40,41,42,43] or (ii) other [32,34,40,43,44]. Cell culture study characteristics and results are summarized in Table 2.

QA of animal studies was carried out using the SYRCLE tool [24] and for cell culture studies using the criteria recommended by the WCRF/UoB framework [22]. The SYRCLE tool resulted in a largely unclear risk of bias for all animal studies (Appendix A, Table A1). The recommended criteria used to evaluate the quality of cell culture studies were vague, resulting in 10 cell culture studies considered to be high quality (5–6) [33,34,35,36,37,40,41,42,43,44] and three studies considered to be moderate quality (3–4) [32,38,39] (Appendix A, Table A2). 

### 3.3. Animal Studies

Animal studies did not compare differences between AS or AI PCa. As such, only animal studies measuring direct androgen outcomes were included in this review. Of the five animal studies included, one study [28] measured the effect of lycopene supplementation on plasma PSA levels, and two studies [27,29] measured whether lycopene or tomato feeding impacted serum testosterone or DHT. Limpens et al. [28] tested the effects of low- and high-dose lycopene supplementation on PSA levels in a xenograft PCa model. After 42 days of daily oral gavage, plasma PSA values did not differ between dietary treatments, suggesting that PSA levels were proportional to tumor size, regardless of dietary intervention. Canene-Adams et al. [27] analyzed the effect of a 10% tomato powder (TP) diet and two different supplemental doses of lycopene (similar dose to the lycopene content of the TP diet and a dose 10-fold higher) on serum testosterone and DHT levels in the Dunning R3327-H transplantable prostate adenocarcinoma model after 18 weeks of tumor growth. None of the interventions had any effect on serum testosterone or DHT levels. Using the same model, Lindshield et al. [29] similarly saw no effect of lycopene supplementation on serum testosterone or DHT levels.

Siler et al. [30] observed reductions in the expression of genes involved in androgen metabolism (SRD5A1) and signaling (cystatin related proteins 1 and 2; prostatic spermine binding protein; prostatic steroid-binding protein C1, C2, and C3; and probasin) in MayLyLu Dunning transplantable tumors with dietary lycopene intake. Only fold-changes in gene expression with no statistical measurements were reported; however, consistent with these results, Wan et al. [31] confirmed that tomato feeding and lycopene supplementation similarly impacted androgen-related gene expression in the prostate of the transgenic mouse model (TRAMP) at early stages of prostate carcinogenesis. Tomato and lycopene diets both decreased the expression of genes related to androgen metabolism (SRD5A2 by tomato and SRD5A1 by lycopene), while the tomato diet reduced the expression of androgen co-regulators paxillin (pxn) and sterol regulatory element binding transcription factor 1 (srebf1).

An albatross plot (Figure 2) was generated for these animal studies to integrate the data, and visual inspection of the plot provided an estimated standardized effect for tomato or lycopene intake on androgen-related outcomes. The effects given were not intended to be precise, as they only provide estimates of the treatment magnitude of effect. The overall SMD of −0.4 (range: 0 to −1.25) represented a reduction in androgen-related outcome measures by exposure to tomato or lycopene. Three studies [27,28,29] showed no effects (assigned *p* = 1) of tomato or lycopene on androgen-related outcomes and as such, cluster at the center (null) line of the plot. The remaining two studies [30,31] showed inverse associations, with SMDs of between −0.5 and −1 and between −1 and −1.5, indicating a reduction in androgen-related outcome measures by exposure to tomato or lycopene. It is important to note that no studies reported an increase in androgen levels or androgen-regulated gene expression with tomato or lycopene exposure, suggesting that neither tomato nor lycopene propagate androgen production or signaling.

### 3.4. Cell Culture Studies

We present the results of cell culture studies here as reported by the original studies with the caveat that care must be taken when interpreting results involving cell culture treatment with lycopene, as antioxidants such as lycopene are extremely labile and are readily oxidized in cell culture [45]. For this reason, lycopene source, purity, storage, delivery vehicle, air and light exposure, and length of time in culture can substantially affect the initial lycopene integrity. These factors vary by study, creating the immediate limitation that the observed results could be due in part to the oxidation products of lycopene, rather than solely to the parent compound.

In general, cell culture studies supported the wide breadth of existing in vivo evidence that tomato and lycopene inhibit PCa tumor growth. Cell culture studies also provide some support for the limited in vivo evidence that tomato and lycopene down-regulate the expression of genes related to androgen signaling and metabolism. The included studies provided mixed evidence to indicate that lycopene interacts with AS and AI PCa cell lines in a differential manner (indirect androgen outcomes) and that lycopene directly interacts with the androgen axis (direct androgen outcomes), as detailed below. Study results provided limited insight into the specific mechanisms by which lycopene might interact with the androgen axis.

#### 3.4.1. Influence of Lycopene on Indirect Androgen Outcomes: Comparison of Androgen-Sensitive vs. Androgen-Insensitive Cell Lines

The limited pool of mechanistic studies makes it difficult to conclude if or how lycopene may directly interact with the androgen axis in PCa to exert growth inhibitory effects. In an effort to glean some insight into this question, it is important to consider how lycopene may impact human PCa cells of differing androgen-responsiveness. To this end, studies that evaluated the differences in outcomes between AS and AI cell lines were included as an indicator of how androgen signaling may influence lycopene activity. Appendix A
Table A3 outlines the differences between each cell line in the included cell culture studies, which may impact the observed results.

(i) Cell Growth

Seven studies [32,33,36,40,41,42,43] evaluated the relationship between PCa cell growth and lycopene treatment. Despite large variation in study methodologies and lycopene doses, all seven studies demonstrated lycopene inhibition of PCa cell growth. All six studies [32,33,36,40,42,43] utilizing human PCa cells lines reported that lycopene treatment inhibited the growth of AS cell lines (LNCaP, LAPC-4), with three studies [33,36,40] reporting greater growth inhibition of an AS cell line (LNCaP) when compared to AI cell lines (C4-2, PC-3, DU145), one study [32] reporting growth inhibition regardless of androgen sensitivity, and two studies [42,43] reporting greater growth inhibition in AI cell lines (PC-3, DU145) than what was observed in an AS cell line (LNCaP). The single study [41] comparing an AS rat-derived PCa cell line (DTE) with its AI daughter cell line (AT-3) reported that lycopene inhibited AI cell growth, but not AS cell growth.

Gong et al. [40] showed that cell growth was reduced by lycopene and its metabolite, apo-10′-lycopenal, in AS LNCaP cells, but not in AI DU145, PC-3, or C4-2 cells. Peternac et al. [36] also found that lycopene inhibited cell proliferation in LNCaP cells and, to a slightly lesser extent, C4-2 cells. Comparably, Linnewiel-Hermoni et al. [33] showed that DHT-induced growth of LNCaP cells and serum-induced (castrate levels of androgens) growth of DU145 and PC-3 cells were inhibited by lycopene treatment, with LNCaP cells exhibiting a more profound response.

Ivanov et al. [32] observed that physiological doses of lycopene (0.2–0.8 µM) resulted in the dose-responsive inhibition of cell proliferation in both LNCaP and PC-3 cells. However, lycopene was observed to exert these effects in each cell line at different mitotic phases, with LNCaP cells mainly undergoing G_0_/G_1_ cell cycle arrest and subsequent apoptosis and PC-3 cells mainly undergoing S and G_2_/M cell cycle arrest without an observed increase in the apoptotic index.

Tang et al. [42] (2005) observed that LNCaP cells resisted apoptosis by high-dose (up to 50 µM) lycopene, while PC-3 and DU145 cells were very responsive to apoptosis by high-dose lycopene. Tang et al. [43] (2011) tested the effects of lycopene on the growth inhibition of PCa cell types with varying androgen sensitivity (LAPC-4, LNCaP, 22Rv1, PC-3, DU145) and found DU145 cells to be the most inhibited by a physiological dose of lycopene (1 µM). Interestingly, PC-3 cells were completely unaffected by lycopene treatment, while LNCaP cells were only marginally affected. AI DU145 cells were not compared with AS LNCaP cells, but both Tang et al. [42] (2005) and Tang et al. [43] (2011) observed that DU145 cells mainly underwent cell cycle arrest at G_0_/G_1_, which contrasts with the S and G_2_/M cell cycle arrest in PC-3 cells observed by Ivanov et al. [32]. The degree of androgen-insensitivity of DU145 cells is greater than that of PC-3 cells, making it possible that the level of androgen insensitivity affects at which stage of the cell cycle lycopene may interfere. More likely, these differing results best serve to highlight the inherent variability of cell culture studies, with different treatment variables potentially impacting outcomes.

Finally, Gunasekera et al. [41] showed significant concentration-dependent decreases of the cell proliferation of malignant, rat-derived, AI AT-3 cells, with lycopene concentrations as low as 0.2 µM and up to 10 µM when compared to the control treatment, with no effect on cell proliferation of AS DTE parent tumor cells by any concentration of lycopene.

An albatross plot summarizing the effects of lycopene on cell growth is presented in Figure 3A. The overall SMD of −2 (range: −0.6–<−2) indicates a reduction in cell growth by lycopene exposure. Two studies [33,36] showed reduced growth with smaller effect estimates (SMDs between −0.5 and −1), two studies [40,43] showed SMDs between −1 and −2, and three studies [32,41,42] showed an SMD < −2.

(ii) Other Outcomes

As introduced in the previous section, Ivanov et al. [32] observed that while physiological doses of lycopene inhibited cell proliferation in both AS LNCaP and AI PC-3 cells at different phases of the cell cycle, lycopene treatment led to similar dose-responsive changes in protein expression, regardless of androgen sensitivity. These effects were not differentially mediated by proteins involved in cell growth pathways in AS and AI cells; protein expression of cyclins D1 and E, cyclin-dependent kinase 2 (CDK2), and Akt phosphorylation were similarly inhibited by lycopene treatment in both cell types.

Tang et al. [43] (2011) attributed the growth inhibitory effects of lycopene in AI DU145 cells when compared to other AS and AI cell types to a correlation with higher levels of insulin-like growth factor-I receptor (IGF-IR) present in cells. The reported order of lycopene leading to the most growth inhibition to the least growth inhibition in cell types was: DU145 > LAPC-4 > 22Rv1 > LNCaP > PC-3; the order of highest to lowest IGF-IR expression in each cell type was: DU145 > PC-3 > LNCaP > 22Rv1 > LAPC-4. However, to determine this correlation, the group used calculated IC_50_ values (reported to be ordered as: LAPC-4 > LNCaP > 22Rv1 > PC-3 > DU145), but did not directly measure lycopene uptake into the cells. Liu et al. [34] (2006) showed lycopene uptake to be much higher in LNCaP cells when compared to PC-3 or DU145 cells (2.5× and 4.5× higher, respectively) at a physiological concentration (1.48 µM). Because lycopene uptake differs between cell lines, using the IC_50_ values calculated from lycopene treatment effects on growth inhibition alone may not be the most appropriate approach. Lycopene was also found to inhibit Akt phosphorylation to a greater extent in DU145 cells (60%) than in LNCaP cells (20%). Because LNCaP cells have a phosphatase and tensin homolog (PTEN) mutation that may lead to enhanced Akt phosphorylation by phosphatidylinositol 3-kinase (PI3K) rather than IGF-IR, the greater inhibition of Akt phosphorylation by lycopene in the DU145 cells may be attributed to lycopene inhibition of the IGF-IR pathway.

Gong et al. [40] showed that lycopene uptake, lycopene cleaving enzyme β-carotene 9′,10′-oxygenase (BCO2) gene expression, and lycopene-induced BCO2 expression were greater in LNCaP cells than in DU145 cells; transfection with either a wild-type (active) or mutant (inactive) BCO2 expression vector led to reduced cell growth in each cell line with or without lycopene treatment. Increased BCO2 expression (wild-type or mutant) also inhibited nuclear factor-κB (NF-κB) luciferase reporter activity by hindering NF-κB p65 subunit nuclear translocation and DNA binding in response to lycopene treatment. Constitutive NF-κB signaling is observed in AI cells lines and is associated with enhanced cell proliferation. These results show that both wild-type (active) BCO2 and mutant (inactive) BCO2 inhibit proliferation, indicating that the anti-proliferative effects of BCO2 are independent of its enzymatic (lycopene cleavage) functions and instead rely on some structural element of BCO2. However, the cellular uptake of lycopene and lycopene-induced expression of BCO2 are dependent on androgen sensitivity, suggesting that lycopene may be less effective at reducing AI cell growth.

Fu et al. [44] compared the effects of lycopene on the methylation and expression of an enzyme involved in detoxification reactions and tumor suppression, glutathione *S*-transferase Pi (GSTP1), in PC-3 and LNCaP cell lines. Treatment of PC-3 cells with 10 µM lycopene significantly reduced levels of GSTP1 promoter methylation, significantly increased the mRNA and protein expression of GSTP1, and significantly decreased the protein expression of DNA methyltransferase 3A (DNMT3A) when compared to control cells. Increasing lycopene treatment to 40 µM showed no additional inhibition of DNMT3A protein expression than the inhibition observed at a dose of 10 µM. Alternatively, LNCaP cells treated with lycopene showed no changes in GSTP1 methylation or expression.

The albatross plot presented in Figure 3B shows a reduction in other androgen-mediated outcomes by lycopene exposure with an SMD of <−3, and one study [40] with an SMD ≈ −2.

#### 3.4.2. Influence of Lycopene on Direct Androgen Outcomes

Liu et al. [34] (2006) reported that lycopene uptake was much higher in AS LNCaP cells when compared to AI PC-3 or DU145 cells. To evaluate whether this higher uptake resulted in direct lycopene binding to the AR, LNCaP cells were transfected with a plasmid containing the ligand-binding domain of L701H, the T877A double mutant, cortisone/cortisol-responsive AR with a broader ligand specificity (AR^ccr^). No direct lycopene binding to the AR occurred, but subcellular fractionation revealed the majority of lycopene to localize within the nuclear membranes and nuclear matrix. Therefore, these data suggest that because lycopene uptake followed the order of AR expression in AS and AI cell lines, lycopene uptake and storage may be mediated by androgen signaling by some mechanism not involving direct binding to the AR.

As discussed in the previous section, Ivanov et al. [32] showed that physiological doses of lycopene decreased cell proliferation in both LNCaP and PC-3 cells at different phases of the cell cycle. These effects were not shown to rely on androgen signaling directly, as the transfection of LNCaP cells with a luciferase-containing androgen response element (ARE) reporter (ARR3-Luc) exhibited no effect of lycopene treatment on androgen-stimulated expression of the gene construct.

Linnewiel-Hermoni et al. [33] showed that DHT-induced growth of LNCaP cells and serum-induced growth of DU145 and PC-3 cells were inhibited by lycopene treatment. To determine whether these effects may be mediated by direct androgen responsiveness, LNCaP cells were transfected with a PSA enhancer luciferase reporter gene construct containing six AREs. Physiological levels of lycopene were not tested, but a high lycopene concentration (8 µM) was found to significantly lower the reporter activity after DHT treatment. Similarly, DHT-induced PSA secretion by LNCaP cells was reported to be ~40% reduced by a more physiological, albeit still high, dose of lycopene (2.5 µM), but statistical analysis did not indicate a significant reduction.

Zhang et al. [39] measured the effect of 0.5–15 µM lycopene on LNCaP cells transfected with a luciferase-containing ARE reporter. The authors reported that lycopene inhibited ARE reporter activity and ARE protein expression in a dose-dependent manner, but failed to include a statistical analysis. Visually, it appears as though 5 µM lycopene acted similarly to 15 µM lycopene, but effects were seen with doses as low as 0.5 µM lycopene. The three studies (Ivanov et al. [32], Linnewiel-Hermoni et al. [33], and Zhang et al. [39]) using luciferase-containing ARE reporters in LNCaP cells show some conflicting evidence. However, it could be insinuated that lycopene effects were directly related to the ARE rather than the AR, mainly at supraphysiological doses of lycopene.

Liu et al. [35] co-cultured primary human prostate cancer stromal (6S) cells with primary normal prostatic epithelial (NPE) cells to determine the effects of DHT on camptothecin (CM)-induced cell death by DNA fragmentation. DHT treatment increased the mRNA expression of IGF-I in 6S cells, which then led to the rescue of CM-induced NPE cell death. Treatment of this co-culture with physiological doses of lycopene (0.3 and 1 µM) inhibited the pro-survival effects of DHT in a dose-responsive manner, potentially due to the administration of lycopene decreasing DHT-induced IGF-I gene expression in 6S cells. Furthermore, lycopene treatment inhibited DHT-induced AR expression in both whole cell lysates and nuclear extracts of 6S cells.

Peternac et al. [36] found that lycopene inhibited cell proliferation in both AS LNCaP and AI C4-2 cell lines. However, these growth effects were not directly related to AR activation, as lycopene had no effect on PSA mRNA or protein levels in either cell line.

Rafi et al. [37] showed that while strictly androgen-regulated genes were largely unaffected in PC-3 cells treated with supraphysiological doses of lycopene (25 µM), some genes within the kallikrein-related peptidase family did show a fold-reduction in expression (klk1, 5, 9, 10, 14). These genes are regulated by members of the steroid hormone family and their expression is typically associated with carcinogenesis, so a slight reduction in gene expression by lycopene treatment may indicate some interference with steroid hormone-regulated gene activation.

Richards et al. [38] measured the effects of low (physiological)- (1 µM) and high-dose (10 µM) lycopene on the PSA secretion of LNCaP cells. While the results showed that PSA protein levels were decreased by about 50% in both treatment groups compared to the control group at all time points, the authors did not report any statistical values or make any comment about the treatments. They also failed to report cell incubation conditions and the mode of delivery of lycopene. Therefore, while these results suggesting that lycopene treatment had direct effects on PSA secretion of LNCaP cells appear promising at first glance, closer inspection revealing the lack of methodological and statistical reporting creates uncertainty when considering the accuracy of the results (refer to Appendix A
Table A2 for study quality assessment).

The albatross plot presented in Figure 3C shows the overall effect estimate as an SMD of −2 (range: 0–<−3), indicating a reduction in direct androgen-mediated outcomes by lycopene exposure. Three studies [32,34,36] showed no effects (assigned *p* = 1) of lycopene on androgen-related outcomes and cluster at the center (null) line of the plot. Three studies [33,35,38] showed SMDs between −0.75 and −1.5, two studies [35,37] showed SMDs between −1.5 and −3, and one study [39] showed an SMD < −3. It is important to note that similar to the effect estimates seen in the animal studies, no cell culture studies reported an increase in androgen-regulated gene activity or expression with lycopene exposure, despite a wide range of effect sizes, suggesting that lycopene has either a neutral or muting impact on androgen signaling.

## 4. Discussion

This systematic review sought to determine whether there was any mechanistic evidence to demonstrate that lycopene directly interacts with the androgen axis during PCa. To our knowledge, this is the first systematic review synthesizing evidence to support the biological plausibility of a role for lycopene interaction with the androgen axis in PCa, a disease state in critical need of the identification of potential therapeutic interventions. The choice to conduct a systematic review was made in accordance with a growing need for the systematic evaluation of preclinical research to determine the strength of the evidence associated with a given topic. In addition, this systematic examination of all available evidence was intended to eliminate the risk of bias incurred by preparing a narrative review. In the absence of validated methods for performing systematic reviews of cell culture and animal studies, we used the guidelines recently set forth by the WCRF/UoB [22]. These guidelines serve as a good starting point for conducting mechanistic systematic reviews and meta-analyses; however, the suggested QA tool provides limited criteria for the critical evaluation of cell culture studies, leading to an inappropriate distribution of high range scores, highlighting a need for validated tools to assess the quality of these studies.

Unfortunately, performing a meta-analysis to statistically analyze the results was not feasible due to the inherent variability in the design and outcomes measured for the included studies. As such, data were grouped according to categories of androgen-related outcomes and graphically presented using albatross plots, a novel method described by Harrison et al. [26] by which to provide estimated SMDs of effects of the available data in the absence of sufficient homogenous data to perform a true meta-analysis. Because insufficient data were available to perform this statistical analysis and because reported outcomes were grouped according to their relationship with the androgen axis, it is important to note that these plots are not intended to provide an exact statistical evaluation of lycopene treatment on the androgen axis. Considering these limitations, the effect estimates shown are largely shifted to the left of the plots, suggesting that tomato or lycopene treatment decreased the effects of androgen signaling or metabolism for almost all outcomes measured. Therefore, while we were not able to determine mechanistically how lycopene interacts with the androgen axis during PCa, we have presented some proposed pathways by which lycopene exerts its anti-androgenic effects (Figure 4). These effects are complex and differ in lycopene uptake and growth pathway activation, depending on model used (animal or cell type), cell metabolism, and androgen signaling and metabolism.

Based on the results of the limited animal studies available for inclusion in this systematic review, there was no evidence that tomato or lycopene intake impact circulating PSA, testosterone, or DHT during PCa. Previous research by our laboratory has shown reductions in serum testosterone by tomato and lycopene intake in a non-PCa rat model [18]. However, while previously thought to be a biomarker for advancing or aggressive PCa, discordance between circulating and intraprostatic levels of androgens, as well as observed adaptive changes in intratumoral steroidogenesis and metabolism, have made serum androgens unreliable markers of disease status [46,47,48]. Instead, changes in intratumoral androgen signaling may be a better indicator of androgen activity than serum testosterone or DHT levels. In accordance with this, there was evidence that tomato and lycopene intake decreased the expression of genes involved in androgen signaling (cystatin-related proteins 1 and 2; prostatic spermine binding protein; prostatic steroid-binding protein C1, C2, and C3; probasin; pxn; and srebf1) and metabolism (SRD5A1 and SRD5A2).

Results from cell culture studies were varied and provided weak mechanistic evidence to demonstrate that lycopene interacts with the androgen axis during PCa. To allow for all possible evidence to be considered, studies comparing differences in cell growth, gene, or protein expression in AS versus AI PCa cell lines were included. Overall, the results showed that cell growth was inhibited by lycopene, regardless of androgen sensitivity. However, the extent to which physiological levels of lycopene inhibit cell growth may be greater in AS cell lines than AI cell lines. In general, AS cell lines more readily accumulate lycopene than AI cell lines [34,40], as shown in Appendix A
Table A3. This difference in lycopene accumulation may contribute to the differences seen in gene expression between the two cell types. AI cells express lower levels of the lycopene metabolizing enzyme, BCO2, which assists in inhibiting tumor promoting NF-κB signaling, independent of its lycopene metabolizing function. However, BCO2 is also inducible by lycopene [40], so the lower lycopene uptake by AI cell types may result in attenuated growth inhibitory effects by BCO2. IGF-1 expression was also shown to be higher in AI cells when compared to AS cells, with lycopene treatment resulting in the inhibition of IGF-1 expression and a resultant decrease in total and nuclear AR expression. Proteomic comparisons between AS and AI cells showed differences in the expression of proteins involved in metabolism, with AI cells exhibiting enhanced glycolysis [49]. Comparisons also revealed AI cells to have decreased poly[ADP-ribose] polymerase 1 (PARP-1) expression when compared to their AS counterparts; PARP-1 plays a role in promoting AR transcriptional activity, with a reduction of PARP-1 correlating with a reduced dependence on androgen signaling in AI cells. Combined, these results suggest that androgen signaling and cellular metabolism interact in PCa cells, with changes in lycopene uptake and metabolism, thereby having the potential to influence androgen signaling.

In addition to its growth inhibitory effects, DNA methylation was inhibited by lycopene in AI cells but not AS cells. DNA hypermethylation is an epigenetic modification that occurs more frequently as PCa progresses [50]. One such example of DNA hypermethylation occurs with GSTP1, a detoxifying enzyme with tumor suppressive activity. GSTP1 promoter hypermethylation and accompanying gene silencing have been consistently detected in more than 90% of PCa cases, with more advanced cases exhibiting higher levels of GSTP1 promoter methylation [51]. Lycopene treatment resulted in reduced GSTP1 methylation and associated suppression of DNA methylating enzyme DNMT3A expression in DU145 cells, but not LNCaP cells. Interestingly, Gong et al. [40] showed that the inhibition of methyltransferase activity resulted in a robust increase of BCO2 expression in all cell lines. This suggests that lycopene may have multiple beneficial effects in both cell types, regardless of androgen sensitivity.

When considering all available studies evaluating lycopene effects on PSA secretion, there is weak evidence, at best, to indicate that lycopene has a direct effect on PSA secretion. A meta-analysis to determine if clinical evidence shows an association between tomato or lycopene intake and PSA levels in humans is currently in progress by our laboratory. Despite a lack of evidence showing a direct impact of lycopene treatment on PSA levels, some evidence exists to suggest that while lycopene does not directly associate with the AR, it may reduce ARE activity. However, this may be at levels higher than the normal physiologic intake of lycopene. Reduction of ARE activity may be associated with lycopene accumulation in the nucleus, which could point to lycopene interference with AR co-regulators and subsequent DNA binding and expression. In addition, both in vivo and in vitro evidence has shown that lycopene influences the expression of genes associated with androgen signaling and metabolism. Combined, these data suggest that because lycopene uptake follows the order of AR expression in AS and AI cell lines, lycopene uptake may be mediated by androgen signaling through mechanisms independent of direct AR binding.

This review is novel in topic and sought to incorporate the use of standardized guidelines recommended for conducting a preclinical systematic review with the addition of quantitative estimates of effects using albatross plots. However, this review presents some limitations. First, few studies mechanistically evaluating the effects of lycopene or tomato intervention on endpoints directly related to androgen status, signaling, or metabolism exist. While cell culture studies measuring endpoints comparing AS and AI cell lines were included for a consideration of the potential differences of lycopene treatment with varying androgen sensitivity, these studies do not directly link lycopene to androgen signaling and must be considered separately from the studies evaluating the effects of lycopene treatment on direct androgen endpoints.

Second, the variability of study design and outcome measurement creates difficultly in comparing animal and cell culture studies. Models used must be carefully evaluated and follow the general progression of human disease. While extremely useful for studying mechanistic outcomes, cell culture models are generally not considered to mimic human disease progression. Cancer, particularly PCa, is considered to be a heterogenous disease. Multiple cell types expressing different mutations are available and considered within this review (e.g., LNCaP, PC-3, DU145), but these single cell lines only represent a small subset of individual tumor phenotypes and do not account for cell interaction with the surrounding tumor microenvironment. In addition, differences between cell culture medium, lycopene dose and delivery method, length of treatment, and laboratory tests conducted may significantly contribute to differences in the overall outcomes. Animal models are a step-up in study design, but only models mimicking the human progression of disease (i.e., tumors originate from the animal) are considered to be high quality models. To this end, only one of the five included studies used a transgenic animal model of PCa (TRAMP), while all others used some form of transplantable tumor model. Furthermore, differences in laboratory tests conducted and outcomes reported for both cell culture and animal studies necessitated the grouping of outcomes by indirect or direct interaction with the androgen axis. As a result, these groupings only served to create estimations of the effects of lycopene treatment on each outcome category, rather than a true statistical evaluation of the effects.

Finally, as previously discussed, lycopene is a potent antioxidant and, as such, is unstable when isolated and exposed to light and air, making it a difficult compound to work with in vitro [45]. Lycopene source, purity, storage, delivery vehicle, air and light exposure, and length of time in culture are all factors that vary by study. These factors can result in oxidation of the parent compound, making study outcomes the result of these oxidized lycopenoids. Studies generally addressed this issue by confirming cellular lycopene uptake, lycopene stability after media culture, or simply by refreshing cell media and treatment daily. However, the labile nature of lycopene and the inherent variability in the study methodology pose major limitations when considering results from cell culture studies.

To address these limitations, future studies should be designed specifically to probe the hypothesis that physiologically relevant doses of lycopene can impact the androgen axis by measuring changes related to androgen activity, signaling, or metabolism. Studies evaluating the effect of lycopene on androgen concentrations, androgen metabolizing enzyme activity, and androgen-regulated gene activity (such as PSA) in animal models, as well as cell lines representative of varying stages of PCa, would result in valuable additions to strengthen the current literature. In the interest of enabling systematic reviews of preclinical research to identify potential mechanisms whereby lycopene can modulate androgen status, future studies should also take care to report a detailed and comprehensive methodology and experimental results.

The reviewed evidence shows that lycopene potentially reduces androgen metabolism and signaling in PCa, thereby reducing the effects of one of the main factors driving PCa growth and progression. While the current pool of research is promising, there is a general lack of preclinical and clinical research relating to the effects or mechanisms of lycopene or other compounds present in tomatoes on androgens, their metabolites, and their downstream effectors at different stages of PCa. Androgen signaling is an important chemotherapeutic target for advanced PCa because intratumoral signaling persists, despite the removal of androgens through ADT. Regular and feasible dietary intake of tomatoes has been shown to reduce the risk for PCa. The mechanisms behind this risk reduction are unclear; however, a reduction of androgen signaling would suggest an important role for tomato and tomato carotenoids at all stages of PCa growth and progression. Therefore, it is essential to identify how simple and widely accepted dietary interventions such as increased tomato intake may act as adjuvant therapies to attenuate the adverse effects of persistent androgen signaling on tumor growth and, as a result, on patient outcome. Future research is needed to fill the large gap that still exists in the literature pertaining to the mechanisms by which tomatoes or lycopene may work to modulate androgen status and androgen signaling during various stages of PCa development and progression.

## Figures and Tables

**Figure 1 nutrients-11-00633-f001:**
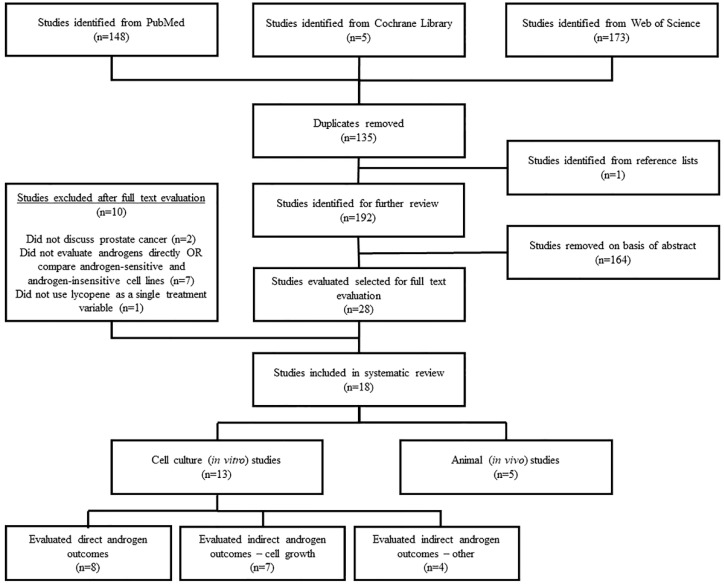
Literature search and study selection flow chart.

**Figure 2 nutrients-11-00633-f002:**
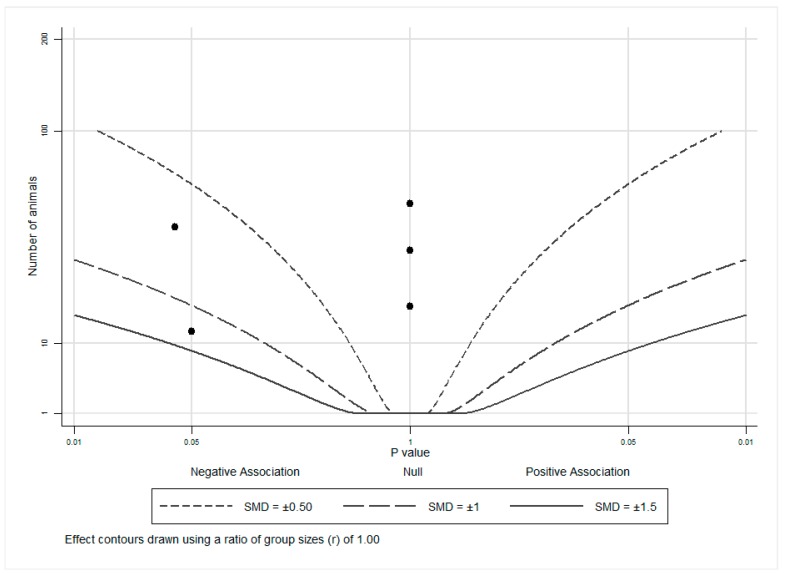
Albatross plot for animal studies. Each point represents a single study, with the effect estimate (represented as a *p*-value), plotted against the total given sample size (*n*) included within each study. Contour lines are standardized mean differences (SMD). *p*-values reported as <0.05 were plotted as 0.05 as a conservative estimate, while non-significant (null) *p*-values were plotted as 1.

**Figure 3 nutrients-11-00633-f003:**
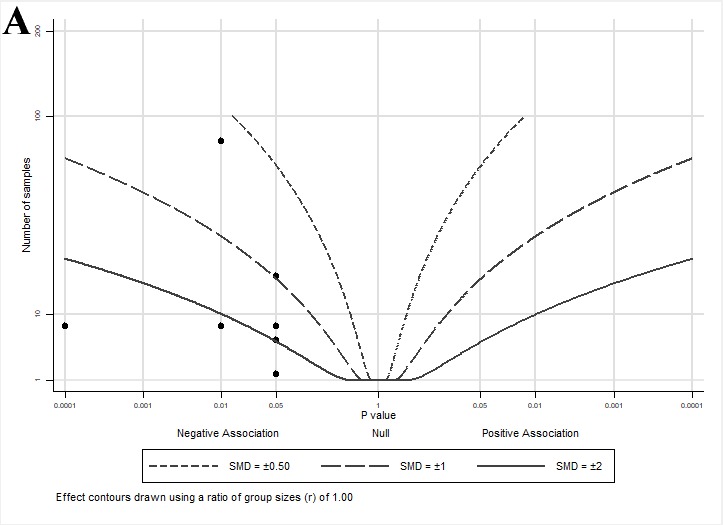
Albatross plots for each outcome of cell culture studies: (**A**) indirect effects (growth); (**B**) indirect effects (other); and (**C**) direct effects. Each point represents a single study, with the effect estimate (represented as a *p*-value), plotted against the total given sample size (*n*) included within each study. Contour lines are standardized mean differences (SMD). *p*-values reported as <0.05 were plotted as 0.05 as a conservative estimate, while non-significant (null) *p*-values were plotted as 1.

**Figure 4 nutrients-11-00633-f004:**
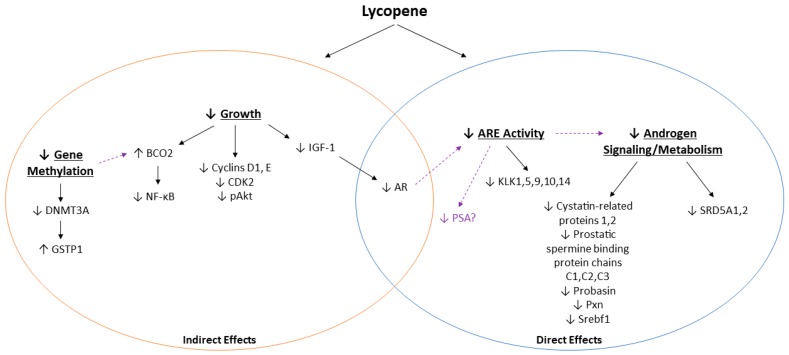
Summary of potential mechanisms by which lycopene may interact with the androgen axis in PCa. Solid lines represent outcomes reported by the reviewed studies, and dashed lines represent potential connections.

**Table 1 nutrients-11-00633-t001:** Characteristics of included animal studies.

Author, Year	Animal Model	Baseline Diet(s)	Dietary Tomato and/or Lycopene Content *	Length of Intervention	Primary Findings
Canene-Adams, 2009 [27]	Dunning R3327-H transplantable tumors (Copenhagen rats)	AIN-93G-based diets fed; *ad libitum*	10% TP (providing 13 nmol lycopene/g diet and resulting in 511 nmol/g serum lycopene), 23 nmol/g diet supplemental lycopene beadlet (252 nmol/g serum lycopene), or 224 nmol/g diet supplemental lycopene beadlet (884 nmol/g serum lycopene)	18 weeks after tumor transplantation	No differences in serum testosterone or DHT between rats fed tomato, lycopene, or control diets
Limpens, 2006 [28]	Xenograft using PC-346C cells (athymic mice)	821077 CRM(P) low (but adequate) vitamin E rodent diet; *ad libitum*	5 or 50 mg/kg BW lycopene (from LycoVit) oral gavage = 0.5–1.5 mg lyco/day	42 days after tumor inoculation	No differences in plasma PSA between mice given control or lycopene gavages (PSA levels were proportional to tumor size regardless of intervention)
Lindshield, 2010 [29]	Dunning R3327-H transplantable tumors (Copenhagen rats)	AIN-93G-based diets; *ad libitum*	250 mg/kg diet supplemental lycopene beadlet = 5 mg lyco/day	18 weeks after tumor transplantation	No differences in serum testosterone or DHT between rats fed lycopene or control diets
Siler, 2004 [30]	MayLyLu Dunning transplantable tumors (Copenhagen rats)	Kliba #2019 with added coconut fat (6%), <5 ppm vitamin E, reduced (but adequate) vitamin A, and devoid of phytosterols; did not indicate if *ad libitum*	200 ppm lycopene (1.02 µM plasma lycopene)	4 weeks on diet prior to tumor transplantation, then 18 additional days	Lycopene supplementation reduced tumor expression levels of *SRD5A1* and androgen-target genes (*cystatin related proteins 1* and *2*; *prostatic spermine binding protein*; *prostatic steroid-binding protein C1*, *C2*, and *C3*; *probasin*) (only fold reductions reported—no other statistical values reported)
Wan, 2014 [31]	Transgenic (TRAMP mice)	AIN-93G -based diets; did not indicate if *ad libitum*	10% TP (providing 384 mg lycopene/kg diet) or 462 mg/kg diet supplemental lycopene beadlet (0.36 µM plasma lycopene) = 3–4 mg lyco/day	4 weeks diet prior to surgery, then 12 additional days	Reduced prostatic expression of genes related to androgen metabolism by tomato feeding (*SRD5A2* (*p* = 0.04), *Pxn* (*p* = 0.04), and *Srebf1* (*p* = 0.05)) and lycopene supplementation (*SRD5A1*, *p* = 0.03)

* All treatments can lead to blood levels of lycopene within a physiological range (~1 µM). Abbreviations: TRAMP (transgenic adenocarcinoma of the mouse prostate); TP (tomato powder); BW (body weight); DHT (dihydrotestosterone); PSA (prostate-specific antigen); SRD5A1 and 2 (5 α-reductase type 1 and 2); Pxn (paxillin); Srebf1 (sterol regulatory element binding transcription factor 1).

**Table 2 nutrients-11-00633-t002:** Characteristics of included cell culture studies.

Author, Year	Cell Line(s)	Culture Conditions *^†^	Lycopene Dose(s) ^δ^	Direct Androgen Outcomes: Primary Findings	Indirect Androgen Outcomes: Primary Findings	
					Growth	Other
Fu, 2014 [44]	LNCaP, PC-3	RPMI1640, 10% FBS	0, 5, 10, 20, 40 µM			10 µM lycopene inhibited GSTP1 methylation (*p* < 0.05), increased *GSTP1* gene expression (*p* < 0.05), and reduced *DNMT3A* gene expression (*p* < 0.01) in PC-3 but not LNCaP cells
Gong, 2016 [40]	LNCaP, C4-2, PC-3, DU145	RPMI1640, 10% FBS	1 µM		1 µM lycopene inhibited growth of LNCaP (*p* < 0.05) but not in C4-2, PC-3, or DU145 cells	1 µM lycopene induced *BCO2* gene expression in LNCaP (*p* < 0.05) but not DU145 cells
Gunasekera, 2007 [41]	AT3, DTE	RD (50% RPMI1640 + 50% DMEM), 2% FBS	0.02, 0.2, 5, 10, 20 µM		0.2 µM lycopene inhibited growth of AT3 (*p* < 0.0001) but not DTE cells	
Ivanov, 2007 [32]	LNCaP, PC-3	RPMI1640 or DMEM, 10% FBS	0.01–10 µM (cell proliferation) 0.2, 0.4 µM (protein expression) 0–100 µM (androgen responsiveness)	Lycopene did not inhibit reporter activity of ARE-Luc transfected LNCaP cells at any concentration (no statistical values reported)	0.2–0.8 µM lycopene inhibited growth of LNCaP and PC-3 cells (*p* < 0.05)	0.2–0.8 µM lycopene inhibited Akt phosphorylation, cyclins D1 and E, and CDK2 in LNCaP and PC-3 cells (no statistical values reported)
Linnewiel-Hermoni, 2015 [33]	LNCaP, PC-3, DU145	RPMI1640 or DMEM, 10% FCS, 10^−9^ DHT (for growth, stripped of steroid hormones prior to treatment)	1–5 µM (cell proliferation) 8 µM (ARE-Luc) 2.5 µM (PSA)	8 µM lycopene inhibited DHT-induced reporter activity of ARE-Luc transfected LNCaP cells (*p* < 0.01); non-significant decrease of DHT-induced PSA secretion by LNCaP cells treated with 2.5 µM lycopene	1–5 µM lycopene inhibited DHT-induced growth of LNCaP cells (*p* < 0.01)	
Liu, 2006 [34]	LNCaP, PC-3, DU145	RPMI1640 or Ham’s F12K or EMEM, 10% FBS	1–1.48 µM	1.48 µM lycopene did not directly bind to the AR (no statistical values reported)		Uptake is highest in LNCaP (*p* < 0.001) with 1.48 µM lycopene compared to PC-3 or DU145 cells
Liu, 2008 [35]	6S, 6S + NPE	DMEM, 5% FBS	0.3, 1 µM	0.3, 1 µM lycopene increased CM-mediated cell death and reduced *IGF-I* gene expression of 6S + NPE cells in the presence of DHT (*p* < 0.01); lycopene reduced DHT-induced total (*p* < 0.05) and nuclear (*p* < 0.01) AR protein expression in 6S cells		
Peternac, 2008 [36]	LNCaP, C4-2	T medium (80% DMEM + 20% Ham’s F12K) + 10% FCS	0.04, 0.4, 4 µg/mL (equivalent to 0.075, 0.75, and 7.5 µM)	Lycopene did not reduce PSA gene or protein expression in LNCaP or C4-2 cells at any concentration	Lycopene inhibited growth of LNCaP (0.04, 0.4, 4 µg/mL) and C4-2 (0.4, 4 µg/mL) cells (*p* < 0.05)	
Rafi, 2013 [37]	PC-3	RPMI1640, 10% FBS	25 µM	25 µM lycopene led to fold reductions of kallikrein peptidase family proteins gene expression in PC-3 cells (only fold reductions reported—no other statistical values reported)		
Richards, 2003 [38]	LNCaP	Did not report cell culture conditions	1, 10 µM	1, 10 µM lycopene appeared to reduce PSA in LNCaP cells (no statistical values reported)		
Tang, 2005 [42]	LNCaP, PC-3, DU145	DMEM + Ham’s F12K, 10% FBS	Up to 50 µM		10–50 µM lycopene more potently inhibited growth of PC-3 and DU145 cells (*p* < 0.01) compared to LNCaP cells	
Tang, 2011 [43]	LNCaP, LAPC-4, PC-3, 22Rv1, DU145	RPMI1640, 10% FBS	1 µM		1 µM lycopene appeared to reduce growth of all cell lines (no statistical values reported)	1 µM lycopene more potently reduced Akt phosphorylation in DU145 (by 60%) than LNCaP (by 20%) cells (no statistical values reported)
Zhang, 2010 [39]	LNCaP	RPMI1640, no other conditions reported	0.5, 5, 10, 15 µM	0.5–15 µM lycopene appeared to reduce reporter activity and ARE protein expression in ARE-Luc transfected LNCaP cells (no statistical values reported)		

* All studies reported standard incubator conditions (5% CO_2_, 37 °C) unless otherwise stated. **^†^** Media does not contain added androgens unless otherwise stated; FBS and FCS supply castrate levels of androgens. ^δ^ Compare to reference of ~1 µM in human plasma. Abbreviations: GSTP1 (glutathione *S*-transferase Pi 1); DNMT3A (DNA methyltransferase 3A); BCO2 (β-carotene 9′,10′-oxygenase 2); ARE (androgen receptor element); Luc (luciferase); CDK2 (cyclin-dependent kinase 2); DHT (dihydrotestosterone); PSA (prostate-specific antigen); AR (androgen receptor); CM (camptothecin); IGF-I (insulin-like growth factor-I).

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
