# Peer review of "Can Lycopene Impact the Androgen Axis in Prostate Cancer?: A Systematic Review of Cell Culture and Animal Studies"

_nutrients, 2019, doi:10.3390/nu11030633_

Reviewer 1 Report

The authors are to be commended for assembling this comprehensive and timely review of the basic research regarding effects of lycopene on the androgen axis and related outcomes relevant to prostate cancer.  The manuscript is clear, well-written, and well-organized, and appropriately outlines the several limitations in both the original experiments and the summarizing of the diverse data and findings.  I have only a few suggestions for clarification or minor improvement.

1. A more accurate title of "Can lycopene......a systematic review of (experimental or laboratory) studies".  This will alert readers interested in data from clinical studies.

2. The search methods and study selection description is clear. I think it would be useful, however, to elaborate in the text about the n=10 (of 28) studies excluded (as explained in the left side text box of Figure 1.

3. Tables 1 and 2 would benefit by the addition of spacing lines between experiments where the column text runs together.  For example, table 1 Dietary tomato column between the first two experiments 1 and 2, and table 2 Direct androgen outcomes between experiments 5-10.

4. I may have missed it, but the authors could elaborate on interpretation of the range of lycopene doses across the experiments, given that there appears to be a range of 2-3 orders of magnitude in the concentrations employed.  For example, did the highest doses result in stronger effects?

5. The Albatross plot figures are useful although their resolution should be improved if possible.

Also, the y-axes of the figures should be relabeled to replace "Participants" with either animals or some metric for the cell culture experiments.

6. Page 19, lines 590-4. Given what this review set out to describe/summarize, it would be useful for the authors to what further studies/experiments would help advance the field and conclusions from where they currently stand (i.e., "Further research is needed.....").  This could also be discuss earlier in this section and not left for the end.

7. Regarding the human evidence for a lycopene-PCa association, a recent pooled analysis of serum lycopene in 11,000 cases and 18,000 controls showing in inverse association for aggressive disease would be a strong, large study to cite in the Introduction (TJ Key et al. AJCN 2015;102:1142-57).

Author Response

Thank you very much for the time you spent reviewing this manuscript, and thank you for providing us with comprehensive comments and useful suggestions. Please see the below responses and summary of modifications made associated with your comments/suggestions:

1.  A more accurate title of "Can lycopene......a systematic review of (experimental or laboratory) studies".  This will alert readers interested in data from clinical studies.

An excellent suggestion. I have changed the title to read: “Can lycopene impact the androgen axis during prostate cancer?: a systematic review of cell culture and animal studies.”

 2. The search methods and study selection description is clear. I think it would be useful, however, to elaborate in the text about the n=10 (of 28) studies excluded (as explained in the left side text box of Figure 1.

A sentence was added (page 4, lines 170-173) to explain the exclusion of the 10 studies to read: “Ten studies were excluded because they did not discuss PCa (n=2), did not evaluate direct androgen outcomes or compare androgen-sensitive vs. androgen-insensitive PCa cell lines (n=7), or did not evaluate lycopene as a single supplement (n=1).

3. Tables 1 and 2 would benefit by the addition of spacing lines between experiments where the column text runs together.  For example, table 1 Dietary tomato column between the first two experiments 1 and 2, and table 2 Direct androgen outcomes between experiments 5-10.

Thank you for that catch. I added some spacing between the mentioned sections, which caused a formatting error with the table footnote that the editors can hopefully fix.

4. I may have missed it, but the authors could elaborate on interpretation of the range of lycopene doses across the experiments, given that there appears to be a range of 2-3 orders of magnitude in the concentrations employed.  For example, did the highest doses result in stronger effects?

This information is mentioned throughout the paper, but in order to clarify whether some of the specific dose ranges mentioned did have dose-responsive effects, the studies were re-checked and updated where applicable. The added information can be found on the following pages/lines:

Page 12: lines 290-291, 310-313

Page 13: line 322, 362-364

Page 14: line 406-407

5. The Albatross plot figures are useful although their resolution should be improved if possible. Also, the y-axes of the figures should be relabeled to replace "Participants" with either animals or some metric for the cell culture experiments.

The figures were edited from “participants” to “animals” or “samples.” The resolution of figure 3 did seem to have been reduced somehow, but nonetheless, all images should be 300dpi now.

6. Page 19, lines 590-4. Given what this review set out to describe/summarize, it would be useful for the authors to what further studies/experiments would help advance the field and conclusions from where they currently stand (i.e., "Further research is needed.....").  This could also be discuss earlier in this section and not left for the end.

In general, more mechanistic and detailed experimental procedures specifically designed to measure androgen-related outcomes would be valuable additions that would help advance this field. To elaborate, a paragraph following the description of the limitations and directly before the descriptions of where our knowledge currently stands was added (pages 19-20, lines 577-585) to read the following:

To address these limitations, future studies should be designed specifically to probe the hypothesis that physiologically relevant doses of lycopene can impact the androgen axis by measuring changes related to androgen activity, signaling, or metabolism. Studies evaluating the effect of lycopene on androgen concentrations, androgen metabolizing enzyme activity, and androgen-regulated gene activity (such as PSA) in animal models as well as cell lines representative of varying stages of PCa would result in valuable additions to strengthen the current literature. In the interest of enabling systematic reviews of preclinical research to identify potential mechanisms whereby lycopene can modulate androgen status, future studies should also take care to report detailed and comprehensive methodology and experimental results.

7.       Regarding the human evidence for a lycopene-PCa association, a recent pooled analysis of serum lycopene in 11,000 cases and 18,000 controls showing in inverse association for aggressive disease would be a strong, large study to cite in the Introduction (TJ Key et al. AJCN 2015;102:1142-57).

Great citation, thank you! That has been incorporated in the introduction (page 2, line 47).

Reviewer 2 Report

This is a comprehensive systematic review of the impact of lycopene on androgen related growth in prostate cancer. The authors reviewed 18 studies and discussed potential mechanisms of the effect of lycopene on prostate cancer cells. There was down-regulation of androgen metabolism and signaling, which may explain in part the preventive effect of lycopene in prostate cancer development and growth.

Author Response

Thank you very much for the time you spent reviewing the manuscript as well as your positive review.